# Structure and Function of Protein Arginine Methyltransferase PRMT7

**DOI:** 10.3390/life11080768

**Published:** 2021-07-30

**Authors:** Levon Halabelian, Dalia Barsyte-Lovejoy

**Affiliations:** 1Structural Genomics Consortium, Temerty Faculty of Medicine, University of Toronto, Toronto, ON M5S 1A8, Canada; l.halabelian@utoronto.ca; 2Department of Pharmacology and Toxicology, University of Toronto, Toronto, ON M5S 1A8, Canada

**Keywords:** protein arginine methylation, PRMT7, epigenetics, cancer, immunity, pluripotency

## Abstract

PRMT7 is a member of the protein arginine methyltransferase (PRMT) family, which methylates a diverse set of substrates. Arginine methylation as a posttranslational modification regulates protein–protein and protein–nucleic acid interactions, and as such, has been implicated in various biological functions. PRMT7 is a unique, evolutionarily conserved PRMT family member that catalyzes the mono-methylation of arginine. The structural features, functional aspects, and compounds that inhibit PRMT7 are discussed here. Several studies have identified physiological substrates of PRMT7 and investigated the substrate methylation outcomes which link PRMT7 activity to the stress response and RNA biology. PRMT7-driven substrate methylation further leads to the biological outcomes of gene expression regulation, cell stemness, stress response, and cancer-associated phenotypes such as cell migration. Furthermore, organismal level phenotypes of PRMT7 deficiency have uncovered roles in muscle cell physiology, B cell biology, immunity, and brain function. This rapidly growing information on PRMT7 function indicates the critical nature of context-dependent functions of PRMT7 and necessitates further investigation of the PRMT7 interaction partners and factors that control PRMT7 expression and levels. Thus, PRMT7 is an important cellular regulator of arginine methylation in health and disease.

## 1. Introduction

Arginine methylation of proteins is a post-translational modification that is introduced by protein arginine methyltransferases (PRMTs). By altering hydrogen bonding, introducing bulk, and some hydrophobicity, arginine methylation can influence protein–protein and protein–nucleic acid interactions, thus playing a role in chromatin, RNA biology, and other phenomena such as phase separation [1,2]. PRMTs regulate normal physiological processes such as myogenesis, embryonic development, and immune system function and play roles in pathologies such as cancer, neurodegeneration, and inflammation [1,2,3,4,5]. Recent knowledge on shared and unique arginine-methylated substrates of PRMTs has shed light on the individual members of the PRMT family. The nine members of the PRMT family are divided into type I represented by PRMT1-4, 6, and 8 that asymmetrically di-methylate the guanidino group of arginine, while type II PRMT5 and 9 engage in symmetric dimethylation of arginines. The sole representative of the type III group is the PRMT7 enzyme that only monomethylates arginine [3,4,6,7,8,9]. The unique structure of this enzyme and the substrate repertoire reflects its function in cells and organisms. This review aims to discuss the recent findings on PRMT7 structure and function, as well as the progress in understanding the roles this enzyme plays in cell biology, disease, and physiological processes.

## 2. Structural Features of PRMT7

### 2.1. Domain Architecture and Evolution

Most PRMTs contain one catalytic seven β strand Rossman fold domain, but require homodimerization to form an active enzyme [10]. However, two family members, PRMT7 and PRMT9, underwent gene duplication in metazoans and thus, contain two tandem domains that fold together, forming a homodimer-like structure. While the N-terminal domain of PRMT7 is catalytically active, the C-terminal domain is considered inactive (see structure discussion below) [11,12] (Figure 1).

Interestingly, outside of metazoans, PRMT7 has been identified in protozoan Kinetoplastida *Trypanosoma* sp. (Figure 2) where this double-domain structure is not conserved, and only one catalytically active domain is present. In the representative species from the animal, fungi, and plant kingdoms, the PRMT7 gene duplication exemplifies the classical double-domain PRMT7 structure. Although PRMT7 has not been described in yeasts, such as *S. cerevisiae*, it is present in several fungi, particularly mold species (Figure 2). Likewise, PRMT7 seems to be absent from the non-vascular plants or even vascular non-seed-bearing plants. Thus, the evolutionary origin and the duplication of PRMT7 warrants further investigation.

### 2.2. Structure

PRMT7 contains two tandem PRMT modules (N and C) that are connected by a 19-residue linker. Each PRMT module in PRMT7 consists of an N-terminal Rossmann fold that is responsible for the cofactor S-adenosylmethionine (SAM)-binding, a C-terminal β-barrel domain for substrate recognition and binding, and a dimerization arm (Figure 3A,B). PRMT7 also contains an additional zinc-finger motif at the junction between the two PRMT modules (Figure 3A), which was shown to lock the module-C in an inactive conformation compared to module-N in the crystal structure of MmPRMT7 in complex with S-adenosylhomocysteine (SAH), the demethylated product of SAM (PDB ID: 4C4A) [11].

Furthermore, several key PRMT signature motifs crucial for SAM/SAH binding are not conserved in module-C [11]. Accordingly, only one SAH molecule was reported to bind to the module-N SAM-binding pocket of the MmPRMT7-SAH complex (PDB ID: 4C4A). Overlay of the SAM-binding domains of modules N and C revealed several residues in module-C, such as D410, P459, F481, and F481 directly overlap with SAM/SAH in the SAM-binding pocket (Figure 4A), indicating that module-C in MmPRMT7 is unable to bind SAM and hence is catalytically inactive.

Recently SGC3027, a highly potent, selective, and cell-active chemical probe for PRMT7 was reported [13]. It represents a cell-permeable prodrug that converts into SGC8158 within the cells. In the crystal structure of MmPRMT7 in complex with SGC8158 (PDB ID: 6OGN), the adenosyl moiety of SGC8158 binds to the SAM-binding pocket of the catalytically active module-N by directly competing with SAM, thus explaining its activity as a SAM-competitive inhibitor (Figure 4B). Moreover, its biphenylmethylamine moiety inserts into an adjacent hydrophobic pocket in the conserved THW motif region, known for substrate arginine coordination in other PRMTs [14,15]. Structural comparison of SGC8158-bound MmPRMT7 with that of TbPRMT7 in complex with H4 peptide (PDB ID: 4M38) shows that only the flexible linker region of SGC8158 overlaps with Arginine sidechain of histone H4 peptide, which may or may not be sufficient to compete with SGC8158 (Figure 4B). Thus, despite the presence of the biphenylmethylamine moiety in the above-mentioned hydrophobic pocket, SGC8158 did not act as a peptide competitive inhibitor.

## 3. Enzyme Function of PRMT7

### 3.1. Regulation, Enzymatic Properties, and Crosstalk with Other PRMTs

Automethylation of PRMT7 R531 was reported to play a role in breast cancer cell migration [16]. Remarkably, the Phosphosite database indicates that human PRMT7 is also monomethylated at R7 and R32, and several ubiquitylation and phosphorylation sites are present. However, the enzymes responsible for these PTMs or their functional outcomes are now known.

Extensive investigations of recombinant PRMT7 in methylation assays indicated that both bacteria and insect cell-expressed recombinant PRMT7 is highly active with a preference for basic arginine-rich substrates such as histones H4 or H2B (KKDGKKRKRSRKESYK peptide) [8,17]. Overall, the PRMT7 enzymatic activity parameters indicate micromolar affinity to SAM and H2B substrate and relatively slow reaction turnover (see references [8,17] for excellent discussion).

One of the more remarkable, recently discovered features of the PRMT7 enzyme is the crosstalk with PRMT5. PRMT5 symmetrically dimethylates H4R3, H2AR3, and H3R8 which are associated with transcriptional repression [18]. Interestingly, PRMT7-directed H3R17 monomethylation drastically increased PRMT5-mediated H4R3 symmetric dimethylation through an allosteric mechanism [19]. Other PRMTs may be similarly affected by PRMT7 monomethylation of neighboring histone arginine residues. Further studies using more complex reaction conditions, substrates, and including PRMT7 binding partners may be able to further address the reaction kinetics of PRMT7, intriguing preference for low reaction temperature, and non-physiological salt concentration preference (discussed in detail in an excellent recent review [20]).

### 3.2. PRMT7 Substrates

The highest enzymatic activity of PRMT7 is found with histone peptides as in vitro substrates. *T. brucei* PRMT7 has also been co-crystalized with the SAM cofactor and H4 peptide (PDB:4M38). Early studies have reported PRMT7-mediated dimethylation of arginines [21,22]; however, subsequent evidence on enzymatic activity, structure, and mutagenesis has unequivocally shown monomethylation activity PRMT7 [6,7,8,9,20]. While histone–peptide substrates have been extensively reported as PRMT7 substrates in vitro, the evidence of PRMT7 dependent histone methylation in cells relies on antibody-based detection in chromatin immunoprecipitation (ChIP) experiments. Several studies noted that PRMT7 modulates H4R3me2s levels at specific loci [23,24,25,26]. In addition, PRMT7 dependent regulation of H2AR3me2s at DNA damage response associated loci was observed [26]. An early study described PRMT5 and PRMT7 regulation of H3R2me2s in association with transcriptional activation Mixed Lineage Leukemia (MLL) complex [27]. In light of the above discussed PRMT7 and PRMT5 crosstalk, it is possible that the observed regulation of H4R3me2s by PRMT7 was an outcome of PRMT7 activating the PRMT5 dimethyltransferase function. Another early study has reported PRMT7-mediated methylation of small nuclear ribonucleoproteins (snRNPs) [21] that was subsequently confirmed in the large-scale mass spectrometry study [28].

Proteomics and antibody-based approaches of methyl-arginine identification have expanded the number of PRMT7 substrates beyond histone proteins (Table 1). However, it should be noted that histone methylation detection by mass spectrometry is challenging as arginine-rich histones are poorly suited for traditional trypsin digestion, and overall differences in modifications at specific genomic loci often fall below detection limits. The most commonly used proteomic approach combining antibody-based enrichment of methyl arginine-containing peptides followed by the mass spectrometry analysis may also introduce an antibody bias. Recently reported antibody-independent methods of methyl arginine detection employing NMR may overcome these limitations [29].

Nevertheless, three recent studies on PRMT7 dependent methylome in mammalian cell lines and *Leishmania* sp. parasite highlight the broad diversity of PRMT7 substrates [13,28,38]. The largest category of proteins enriched in the PRMT7 methylated hits was RNA binding and metabolism-associated proteins, a finding that was consistent in all three studies. Interestingly, these studies also enabled elucidation of the preferred methylation motif of PRMT7. Numerous in vitro experiments have attributed the RXR motif as highly methylated by PRMT7 [8,39]. However, the proteomic studies indicate an overall preference for methyl arginine to reside in glycine-rich regions [28,38], whilst in mammalian cells, there is a slight preference for proline to precede the methyl arginine [28].

### 3.3. Inhibitor Compounds for PRMT7

The discovery of potent and selective inhibitors for PRMT enzymes has enabled experimental approaches to specifically address the catalytic functions of these enzymes and facilitated therapeutic development [17,40]. One of the first compounds described as a dual inhibitor for PRMT5 and PRMT7, DS-437 was based on the SAM cofactor design. Although relatively potent in vitro (6 μM), the compound required high cellular concentrations to inhibit PRMT5 activity in cells, and specific PRMT7 activity was not addressed [41]. Further exploration and optimization of compounds occupying the SAM binding pocket of PRMT7 yielded extremely potent (2.5 nM) in vitro compounds selective for PRMT7 over other PRMT family members and other methyltransferases (Figure 4B). Due to the poor cellular permeability of the chemical SAM scaffold, a prodrug strategy was employed to generate cell-active inhibitors. SGC3027 is a prodrug compound that, upon reduction by abundant cellular reductases, releases the active component of SGC8158. Importantly the negative control compound is also available to ensure meaningful experimental data [13]. SGC3027 compound has been demonstrated to inhibit PRMT7-dependent methylation of HSP70 and other protein substrates in cells [13,28].

## 4. Cellular Roles of PRMT7

### 4.1. The Role of PRMT7 in Gene Expression and Genome Maintenance

PRMT7 methylates histones and results in gene transcription regulation. Repressive H4R3me1 and H4R3me2s marks were associated with PRMT7 activity on the BCL6 promoter, although the latter could be due to the allosteric activation of PRMT5. PRMT7 regulates B cell development, and overexpression in the B cell lineage cell lines resulted in lower BCL6 levels and higher H4R3me2s at the promoter of *Bcl6* [26]. Another study found that PRMT7 dimethylated H2AR3, and H4R3 were enriched on DNA repair genes. Knockdown of PRMT7 upregulated the expression of multiple transcripts involved in DNA repair, including ALKBH5, APEX2, POLD1, and POLD2 [25]. Expression of these genes, especially DNA polymerase (POLD1), could mediate the sensitivity to DNA damage conferred by PRMT7 [25]. PRMT7 can also antagonize the action of the MLL methyltransferase complex. MLL4 is the H3K4 methyltransferase that plays a role in cellular differentiation. Knockdown of PRMT7 enhanced the levels of H3K4me3, decreased H4R3me1, and increased the expression of MLL4 target genes, promoting neuronal differentiation [24]. The factors that recruit PRMT7 to these complexes and gene loci are not always clear, as is the context of other histone modifications that may influence PRMT7-driven arginine methylation.

A study investigating the function of PRMT7 in muscle physiology determined that PRMT7 knockdown in C2C12 cells resulted in a decrease of H4R3me2s on the promoters of several genes, including *Dnmt3b* and *Cdkn1a*. However, in PRMT7-deficient cells, the activating mark H3K4me3 was decreased at the *Dnmt3b* promoter, while it was increased at the *Cdnk1a* promoter, thus, correlating with reduced expression of *Dnmt3b* and increased Cdkn1a mRNA levels that resulted in premature senescence [23]. Another study examining epigenetic regulation of imprinted genes identified CTCFL/BORIS, a paralog of CTCF, as PRMT7 binding partners. PRMT7 was recruited to imprinting control regions during embryonic male germ cell development. This resulted in the symmetric dimethylation of H4R3 at nearby nucleosomes, thereby facilitating the recruitment of the de novo DNA methyltransferases 3 (DNMT3a/b). DNA methylation of the imprinting control region determines the parental specific expression of *Igf2* in male germ cells [42]. Such tissue-specific roles of PRMT7 through the interaction with tissue-restricted binding partners may prove to be more widespread than previously thought, as PRMT7 knockout studies indicate distinct functional outcomes in distinct cell types (see below).

### 4.2. Regulation of Pluripotency, Cell Differentiation, and Senescence

The balance between cellular states is controlled by the intricate orchestration of cellular signaling and transcription factors. PRMT7 is highly expressed in pluripotent cells [43]. Examination of candidate reprogramming factors in mammalian oocytes determined that PRMT7 protein levels were the highest in the pluripotent oocyte and changed substantially during mouse embryogenesis. In addition, the authors found that PRMT7 replaced SOX2 as one of the Yamanaka factors in generating induced pluripotent stem cells (iPSC) from mouse embryonic fibroblasts [44].

By modifying H4R3me2s, PRMT7 repressed the *miR-24-2* gene cluster that downregulates the expression of *Oct4, Nanog, Klf4,* and *c-Myc*. These miRNAs also targeted the 3′UTR of their repressor gene *Prmt7* thus forming a double-negative feedback loop where miR-24-3p/miR24-2-5p downregulates PRMT7 and vice versa to control Oct4, Nanog, Klf4, and c-Myc in pluripotency [45]. PRMT7-mediated repression of another miRNA cluster, miR-221-3p and miR-221-5p, also plays a critical role in pluripotency factor Oct4, Nanog, and Sox2 levels and mouse embryonic stem cell stemness [46]. Remarkably PRMT7, together with PRMT5, regulates mouse embryonic development from the 2-cell to 4-cell stages and plays a role in early human embryonic developmental arrest [47,48].

Several reports highlight the role of PRMT7 in normal tissue homeostasis. PRMT7 is preferentially expressed in injury-activated muscle satellite cells and is required for muscle regeneration. As mentioned above, PRMT7 regulates histone methylation and thus p21CIP and DNMT3b expression, leading to cell-cycle arrest and premature cellular senescence, consequently resulting in a deficiency of regenerating myofibers, a reduced pool of PAX7-positive cells, and a failure of satellite cells to self-renew [23]. Interestingly another mouse knockout study demonstrated that PRMT7-deficient muscle exhibit decreased oxidative metabolism, which is associated with reduced expression of PGC-1α, a critical regulator of the mitochondria. Changes in muscle structure and fiber type were attributed to PGC-1α that in turn was regulated by p38MAPK. The authors provided a link between PRMT7 methylation of p38 mitogen-activated protein kinase (p38MAPK), which activates Activating Transcription Factor 2 (ATF2), an upstream transcriptional regulator for PGC-1α [37]. The cellular senescence phenotype observed in PRMT7 deficiency was also found in mouse embryonic fibroblasts where premature senescence coincided with reduced levels of sonic hedgehog (SHH) pathway regulator GLI2. The authors have shown that PRMT7 promotes SHH signaling via GLI2 methylation regulating the localization of GLI2 [33]. This study is especially important in light of the complex regulation of GLI1 and GLI2 by PRMT1, PRMT5, and PRMT7 that controls cell senescence, self-renewal with potentially far-reaching implications in pluripotency and cancer-initiating cell biology [49].

Conditional knockout of PRMT7 in the B-cell lineage resulted in impaired B-cell differentiation and hyperplasia of the germinal center. In contrast, over-expression of PRMT7 triggered an increase in BCL6 in germinal center-derived B-cell lines. Thus, PRMT7 overexpression impairs lymphoid differentiation, and normal PRMT7 function is needed for B cell development [26]. Interestingly, PRMT7 also plays a role in adipogenesis by controlling C/EBP-β activity or PPAR-γ2 expression [50]. Overall, PRMT7 plays an essential role in regulating the cellular states of pluripotency, differentiation, senescence, and the epithelial–mesenchymal transition discussed below.

### 4.3. PRMT7 and Stress Response

One of the earliest functional descriptions of PRMT7 came from studies in Chinese hamster cell line DC-3F, linking low levels of PRMT7 to resistance to the topoisomerase II inhibitors 9-OH-ellipticine, etoposide, and cisplatin [51,52]. Genomic linkage studies indicated that PRMT7 resides in the susceptibility to etoposide-induced cytotoxicity loci [53]. In contrast, another study showed that downregulation of PRMT7 isoforms in DC-3F hamster cells was associated with increased sensitivity to the topoisomerase inhibitor camptothecin [54]. Subsequent work by Karkhanis et al. demonstrated that PRMT7 regulates DNA damage response genes and thus the sensitivity to DNA damage [25]. In addition to the above-mentioned regulation of *POLD1*, PRMT7 interacts with BRG1 and BAF, SWI/SNF chromatin remodeling subunits to regulate methylation H2AR3 and H4R3 and suppress DNA repair gene expression, subsequently resulting in the sensitization to the DNA damage stress [25].

Several other regulators involved in cellular stress response have been associated with PRMT7 function. PRMT7 interacts with and can methylate eukaryotic translation initiation factor 2 alpha (EIF2S1) at R55 and in neighboring arginine [31]. Various stresses can result in EIF2S1 phosphorylation, translational shutdown, and unfolded protein response [55]. Haghandish et al. showed a regulatory interplay between EIF2S1 arginine methylation by PRMT7 and the S51 phosphorylation status of eIF2α. Upon translational stress, EIF2S1 is phosphorylated, and PRMT7 is required for EIF2S1-dependent stress granule formation that sequesters transcripts, translational machinery, and initiates a protective response program [31]. Interestingly the eukaryotic elongation factor 2 (EEF2) and stress granule protein G3BP2 were also reported methylated by PRMT7 [32,56].

PRMT7 methylates HSP70 protein family members HSPA1A/B, 6, 8 [13,57]. HSP70 protein chaperones are critical for folding new proteins, maintaining protein homeostasis, and are highly upregulated in response to various stressors [58]. Methylation of HSP70 R469 by PRMT7 facilitated the correct substrate refolding after heat shock and regulated the magnitude of the stress granule response after proteostasis perturbations due to proteasome inhibition. These protective features of methylated HSP70 were associated with higher resistance of wild-type cells to proteasome inhibition when compared to PRMT7 knockout cells [13].

It is possible that other PRMT7 substrates are also involved in stress response, and incidentally, the largest category of PRMT7 methylated proteins are RNA binding proteins [13,28,38] that do play a variety of roles in the stress response [59]. Interestingly the prominent role of PRMT7 in muscle cell physiology (see above) may be linked to extensive regulation of proteostasis in this tissue. Evolutionary adaptation to stress may underlie such phenomena as the noted preference of PRMT7 enzymatic activity to lower temperatures than 37 °C [8,60]. The role of PRMT7 in stress and adaptation would be consistent with highly context-dependent PRMT7 phenotypes in cells and organisms.

## 5. Connection to Disease and Organismal Phenotypes

### 5.1. Knockout Phenotypes

Several studies have addressed the organismal role of PRMT7. PRMT7-knockout mice generated by a gene-trap approach displayed significantly reduced body size, reduced weight, and shortened fifth metatarsals. These mice were subviable with surviving adult mice exhibiting increased fat mass, limb and bone anomalies, such as reduced bone mineral content [61]. The subviable nature of PRMT7 knockouts was also noted in another study that subsequently derived B cell-specific PRMT7 knockouts [26]. In this context, PRMT7 loss did not result in changes in frequency and number of early B cell subpopulations but led to decreased mature marginal zone B cells, increased follicular B cells, and promoted germinal center formation. In addition to the aforementioned repressive histone methylation on the *Bcl6* promoter by PRMT7, the authors provided clear links to downstream gene expression programs that involved integrin-mediated cell adhesion. Another exciting avenue discussed in this study was PRMT7 association with the misregulation of DNA damage response that may play an essential role in resting B cells [26].

The role of PRMT7 in muscle physiology has been investigated by several groups. Here, whole-body knockouts of PRMT7 were found viable, possibly indicating that the mouse strain context is important. The mutant animals had decreased muscle regeneration after injury due to the loss of a stem cell population of satellite cells, see above for mechanistic discussion [23]. Jeong et al. noted age-associated obesity in PRMT7-deficient mice and changes in overall muscle structure with the shift from fast-twitch glycolytic fibers to slow-twitch oxidative phosphorylation dependent fibers [37]. These studies indicate the important role PRMT7 plays in normal adult muscle function.

PRMT7 knockout mouse brain dentate granule cells displayed increased firing frequency attributed to enhanced NALCN, the resting membrane potential regulator, and overall hyperexcitability in the knockout brain granule cells. PRMT7 methylates a highly conserved Arg1653 of the NALCN, leading to NALCN Ser1652 phosphorylation, NALCN inhibition, and reduced neuronal excitability [34]. It will be interesting to see how this PRMT7 control of intrinsic excitability in hippocampal neurons translates to organismal phenotype.

Interestingly PRMT7 knockout zebrafish were more resistant to spring viremia of carp virus (SVCV) and grass carp reovirus (GCRV) infections and exhibited enhanced expression of critical antiviral genes. Thus, PRMT7 negatively regulates antiviral responses in zebrafish that involve retinoic acid-inducible gene 1 (RIG1) [62]. A recent study in mice also indicates increased resistance to viral infection-induced lethality in PRMT7 knockout animals [35]. These experiments indicate an important organismal role of PRMT7 in innate immunity regulation that may extend to other organisms (also discussed below).

### 5.2. Human Syndromes

A study on human genetic developmental disorders identified PRMT7 mutations as responsible for a phenotype that phenocopies pseudohypoparathyroidism with mild intellectual disability, obesity, and symmetrical shortening of the digits, posterior metacarpals, and metatarsals. Sexually dimorphic features, including changes in bone mineral content, bone mineral density, and fifth metacarpal length in females were also noted [61]. Overall, several studies established the links between PRMT7 mutations and OMIM phenotype 617157 of Short Stature, Brachydactyly, Intellectual Developmental Disability and Seizures (SBIDDS) [61,63,64,65,66]. Further research is needed to draw parallels between this phenotype and the mouse studies to better understand PRMT7 function and evolution.

### 5.3. Cancer

Several studies report overexpression of PRMT7 in cancer. Although the Cancer Dependency Map [67] does not indicate that PRMT7 is essential for cell survival under normal growth conditions, it is possible that in specific contexts, PRMT7 plays a role in tumorigenesis. One of these contexts is the epithelial–mesenchymal transition (EMT) which occurs during embryonic development, tissue regeneration, organ fibrosis, and cancer metastasis and survival. In EMT, epithelial cells lose polarity, cell-cell junctions, epithelial markers, and gain cell motility, a spindle-cell shape, and mesenchymal markers [68]. In breast carcinoma cells, increased PRMT7-mediated EMT and metastasis by losing E-cadherin expression due to altered histone methylation, specifically elevated H4R3me2s levels [69]. Baldwin et al. highlighted that PRMT7 is overexpressed in basal breast cancer cells, and the knockdown of PRMT7 reduces cell motility and invasion. Conversely, overexpression of PRMT7 in epithelial breast cancer cells promotes cell invasion by upregulating the MMP9 matrix metalloproteinase that is responsible for the degradation of the extracellular matrix enabling cancer cells to invade tissues [70]. PRMT7 automethylation itself also seems to play a role in breast cancer metastasis [16]. The allosteric regulation between PRMT7 and PRMT5 should be considered, especially in breast cancer, where the role of PRMT5 in cancer-initiating cells and disease progression has been well established [71,72].

PRMT7-dependent methylation of R21 in mitochondrial ribosomal protein S23 MRPS23 accelerated the polyubiquitin-dependent degradation of MRPS23. MRPS23 degradation inhibited oxidative phosphorylation and increased mitochondrial reactive oxygen species (ROS) levels, consequently increasing breast cancer cell invasion and metastasis. As low levels of MRPS23 result in breast cancer cell survival through regulating oxidative phosphorylation, PRMT7 overexpression inhibited oxidative phosphorylation and increased breast cancer cell invasion [36]. PRMT7 overexpression also promoted the invasion and colony formation in lung cancer, and the authors of this study found that PRMT7 interacted with mitochondria localized HSP70 family member HSPA5 and elongation factor 2 EEF2 [73]. Thus, overall, the evidence indicates that PRMT7 plays an important role in tumorigenesis by regulating the cellular differentiation states.

### 5.4. Immunity and Infection

Arginine methylation regulates immune cell function and antiviral response [1,40]. The above-mentioned study on zebrafish PRMT7 downregulating the viral response genes and conferring susceptibility to infection indicated that PRMT7 plays an important role in the immune response [62].

Recent work demonstrated that PRMT7 methylates mitochondrial antiviral-signaling protein (MAVS) on R52 and attenuates MAVS binding to its partner proteins TRIM31 and RIG1 that is key to the downstream antiviral signaling [51]. Viral component binding to RIG-I and melanoma differentiation-associated gene 5 (MDA5) induces their interaction with MAVS via N-terminal caspase recruitment domains (CARDs). The activated MAVS CARD rapidly forms aggregates converting other MAVS on the mitochondrial outer membrane into prion-like aggregates [74]. PRMT7-mediated methylation affects the aggregation of MAVS that is essential for the biological functions of MAVS. Consequently, the PRMT7 inhibitor SGC3027 enhanced interferon signaling, *Ifnb1*, *Isg56*, and *Cxcl10* gene expression downstream of MAVS, at the same time reducing the viral titers from infected cells. Interestingly, the authors show that the loss of one PRMT7 allele protects mice from viral infections [35].

Thus, both studies in mice and zebrafish above indicate that PRMT7 confers susceptibility to viral infections and enhances the immune response. The evolutionary origins of this phenomenon and its biological significance warrant further investigation. Given the complexity of the immune and antiviral responses, as well as links to B cell biology, it will be interesting to see if PRMT7 plays a role in autoimmunity and viral mimicry in cancer.

## 6. Conclusions and Outlook

A wide variety of biological processes associated with PRMT7 function are consistent with the broad range of proteins methylated by PRMT7 in cells. This repertoire will undoubtedly be further expanded in the course of future research. In light of the emerging pattern of context-dependent function of PRMT7, it will be interesting to explore tissue, cell state, or stimulus-specific roles of PRMT7. So far, the organismal knockouts of PRMT7 have provided a wealth of phenotypic information of which only some are mechanistically accounted for, such as the role of PRMT7 in muscle physiology or B cell biology. However, further studies on the processes underlying neuronal or bone development as well as metabolic phenotypes of PRMT7 knockouts are needed. Concurrently understanding PRMT7 expression patterns and the transcription factors that determine cell-specific or stimulus-driven PRMT7 regulation will enable deeper mechanistic knowledge. And as indicated above, the context-specific function of PRMT7 may stem from its interacting proteins. Several studies have begun to address binding partner proteins of PRMT7 through co-immunoprecipitation or proximity biotinylation techniques, where the former has further delved into the biological outcome of such interactions identifying EIF2S1 methylation and stress response regulation [40]. Such mechanistic research will be necessary to elucidate the significance of PRMT7-driven methylation. Thus, addressing PRMT7 expression pattern, protein stability, substrates, and interaction partners and interplay with other PRMT family members will enable a deeper understanding of PRMT7 function.

## Figures and Tables

**Figure 1 life-11-00768-f001:**
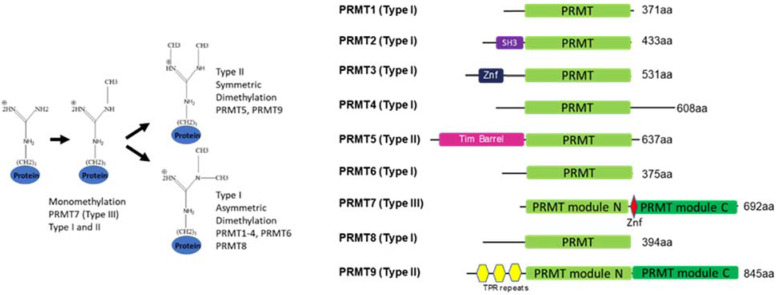
General overview of protein arginine methylation modes (**left**) and PRMT family domain structure (**right**).

**Figure 2 life-11-00768-f002:**
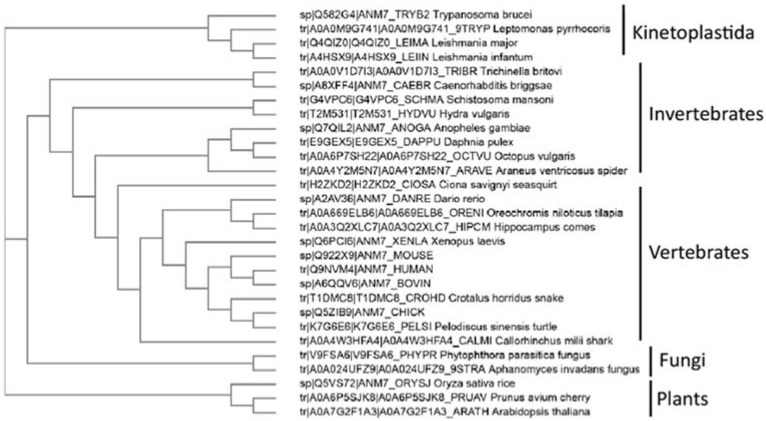
Evolutionary analysis of known PRMT7 proteins. Known PRMT7 sequences from the UniProt database were aligned, and cladogram-rendered using ClustalW2 and simple phylogeny software (EBI).

**Figure 3 life-11-00768-f003:**
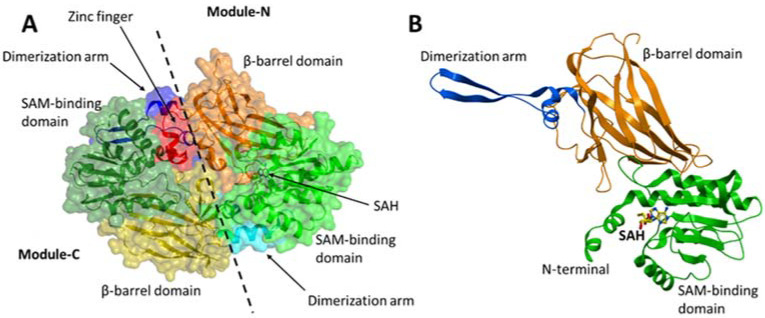
Crystal structure of MmPRMT7 in the complex with SAH (PDB ID: 4C4A). (**A**) Overall structure of MmPRMT7 is shown in the surface representation color-coded according to its domain boundaries. The two catalytic modules (N and C) in MmPRMT7 are divided by a dashed line. (**B**) Module-N of MmPRMT7 is shown as cartoon representation in green (SAM-binding domain), orange (β-barrel domain) and blue (dimerization arm), and SAH in yellow sticks.

**Figure 4 life-11-00768-f004:**
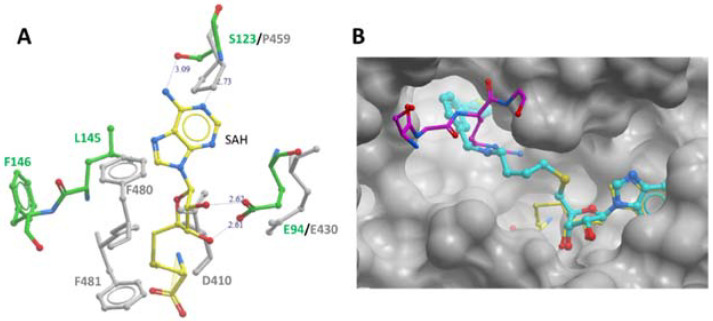
PRMT7 SAM-binding pocket comparison and occupancy by the inhibitory compound SGC 8158. (**A**) Comparison of SAM-binding domains of modules N and C. Overlay of module-N (in green) and module-C (in grey) SAM-binding domains of MmPRMT7 in complex with SAH (yellow). (**B**) Close-up view of module-N SAM-binding domain in complex with SGC8158 chemical probe (cyan) (PDB ID: 6OGN). The SAM-binding domains for both MmPRMT7-SGC8158 and TbPRMT7-SAH in complex with H4 peptide (PDB ID: 4M38) were superimposed to show the SGC8158 binding mode relative to SAH (yellow) and H4 peptide (magenta).

**Table 1 life-11-00768-t001:** Selected PRMT7 substrates in cells.

Substrate	R Methylation Sites	Function, Disease Relevance, Reference
DVL3	271, 342, 614	DVL3 localization, wnt signaling, cancer [30]
EIF2S1 (EIF2 alpha)	55	Translation arrest, stress granule regulation, [31]
G3BP2	432, 438, 452, 468	Wnt signaling, cancer, [32]
GLI2	225/227	Cell senescence, [33]
Histone H4, H2A	H4R3, H2AR3	Gene expression, [23,24,25,26,27]
HNRNPA1	194, 206, 218, 225	Splicing, [28]
HSP70	469	Stress response, [13]
NALCN	1653	Neuronal excitability, [34]
MAVS	52	Viral infection, [35]
MRPS23	21	Oxidative phosphorylation, cell invasion, cancer [36]
P38MAPK	70	Myoblast differentiation, [37]
PRMT7	531	Cell migration, cancer [16]

## Data Availability

Not applicable.

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
