# Peer review of "Structure and Function of Protein Arginine Methyltransferase PRMT7"

_life, 2021, doi:10.3390/life11080768_

Round 1
Reviewer 1 Report
This manuscript is a comprehensive review on PRMT7, an arginine methyltransferase that is unique from structural and enzymatic points of view, and whose biological functions are becoming clear with many of the studies cited. This review is well-written and provides lots of interesting, up-to-date detail. Below are some suggestions and minor edits to improve this work prior to publication.
Line 27: Arginine methylation of proteins is a post-translational process and not post-transcriptional.
Line 35: Rather than citing reviews, perhaps the author may consider citing primary literature.
Line 67: S-adenosylmethionine is one word similar to S-adenosylhomocystene.
Line 96: “the adenosyl moiety of SGC8158 binds to the SAM-binding pocket of the catalytically active module-N by directly competing with SAM”; remove “of”
Line 101: Perhaps the author should explain why SGC8158 does not act as a peptide competitive inhibitor. Does this result suggest a specific enzymatic mechanism? What is the significance in making this point?
Lines 118, 119, 172, 189, 193, 229, 300, 336, 346, and 368: PRMT7-directed, PRMT5-mediated (add hyphens)
Line 128: T. brucei should be italicized.
Lines 136 and 382: should read “In light of…”
Line 176: This sentence implies that the monomethyltransferase PRMT7 is responsible for depositing H4R3me2s. Is that really the case?
Line 253: PRMT7 instead of PRMT
Line 307: “leading to NALCN SER1652…”
Line 372: “ Interestingly,…”
Author Response
Dear Reviewer,
We would like to thank you for your time and helpful comments as well as suggestions. We considered them very carefully, added the requested information, and made changes to the manuscript that will facilitate scientific communication, interpretation and will provide a broad and inspiring overview of PRMT7. Below, as outlined in the following pages, is the detailed list addressing the suggestions and concerns. The changes in the manuscript text are indicated. We hope that the revisions will meet your expectations and look forward to your response.
Yours sincerely,
Dalia
Reviewer 1
This manuscript is a comprehensive review on PRMT7, an arginine methyltransferase that is unique from structural and enzymatic points of view, and whose biological functions are becoming clear with many of the studies cited. This review is well-written and provides lots of interesting, up-to-date detail. Below are some suggestions and minor edits to improve this work prior to publication.
Line 27: Arginine methylation of proteins is a post-translational process and not post-transcriptional.
We thank the reviewer for catching this embarrassing mistake. It was corrected.
Line 35: Rather than citing reviews, perhaps the author may consider citing primary literature.
The intent was to cite reviews in statements that were general to PRMTs. However, we realize that the reviews covering the statement of PRMT7 monomethylation activity were too general; thus, on line 35 we have added additional primary literature.
Line 67: S-adenosylmethionine is one word similar to S-adenosylhomocystene.
We thank the reviewer for pointing this out. It has been corrected.
Line 96: “the adenosyl moiety of SGC8158 binds to the SAM-binding pocket of the catalytically active module-N by directly competing with SAM”; remove “of”
The suggested edit was made, and currently, the sentence reads as: “In the crystal structure of MmPRMT7 in complex with SGC8158 (PDB ID: 6OGN), the adenosyl moiety of SGC8158 binds to the SAM-binding pocket of the catalytically active module-N by directly competing with SAM, thus explaining its activity as a SAM-competitive inhibitor”.
Line 101: Perhaps the author should explain why SGC8158 does not act as a peptide competitive inhibitor. Does this result suggest a specific enzymatic mechanism? What is the significance in making this point?
We thank the reviewer for requesting clarification. In our previous work on PRMT inhibitors, we noted that the relationship between substrate/cofactor pocket occupancy and competitive behaviour is quite complex. Here we updated the text and Figure 4B to clarify the inhibitor occupancy and competitive behaviour. “Structural comparison of SGC8158-bound MmPRMT7 with that of TbPRMT7 in complex with H4 peptide (PDB ID: 4M38) shows that only the flexible linker region of SGC8158 overlaps with Arginine sidechain of histone H4 peptide, which may or may not be sufficient to compete with SGC8158 (Figure 4B). Thus, despite the presence of the bi-phenylmethylamine moiety in the above-mentioned hydrophobic pocket, SGC8158 did not act as a peptide competitive inhibitor.”
Lines 118, 119, 172, 189, 193, 229, 300, 336, 346, and 368: PRMT7-directed, PRMT5-mediated (add hyphens)
Corrected
Line 128: T. brucei should be italicized.
Corrected
Lines 136 and 382: should read “In light of…”
Corrected
Line 176: This sentence implies that the monomethyltransferase PRMT7 is responsible for depositing H4R3me2s. Is that really the case?
We thank the reviewer for pointing this out. The sentences were corrected to: “PRMT7 methylates histones resulting in gene transcription regulation. Repressive H4R3me1 and H4R3me2s marks were associated with PRMT7 activity on the BCL6 promoter, although the latter could be due to the allosteric activation of PRMT5. PRMT7 regulates B cell development, and overexpression in B cell lineage cell lines resulted in lower BCL6 levels and higher H4R3me2s at the promoter of Bcl6.”
Line 253: PRMT7 instead of PRMT
Corrected
Line 307: “leading to NALCN SER1652…”
Corrected
Line 372: “ Interestingly,…”
Corrected

Reviewer 2 Report
Please see the attachment

Author Response
Dear Reviewer,
We would like to thank you for your time and helpful comments as well as suggestions. We considered them very carefully, added the requested information, and made changes to the manuscript that will facilitate scientific communication, interpretation and will provide a broad and inspiring overview of PRMT7. Below, as outlined in the following pages, is the detailed list addressing the suggestions and concerns. The changes in the manuscript text are indicated. We hope that the revisions will meet your expectations and look forward to your response.
Yours sincerely,
Dalia
Reviewer 2
Manuscript ID: life-1301541
This is an interesting and well-written review on the structural and functional properties of PRMT7 that will benefit the PKMT readers. I include some suggestions/additions that may improve the manuscript.
Major comments:
1) There is evidence that PRMT7 may also methylate the Small nuclear ribonucleoproteins (snRNPs). The authors may want to discuss them as non-histone substrates.
We thank the reviewer for pointing this out. The following sentence was added: “Another early study has reported PRMT7-mediated methylation of small nuclear ribonucleoproteins (snRNPs) [25] that was subsequently confirmed in the large-scale mass spectrometry study [26].”
2) Additionally, they may also want to describe the PRMT7’s role in regulating the cell’s DNA repair machinery.
We thank the reviewer for this comment. We have added the following information to section 4.3 where we discuss the stress response. The Karkhanis et al study is extensively mentioned throughout the manuscript. The sentence added:
In addition to the above-mentioned regulation of POLD1, PRMT7 interacts with BRG1 and BAF, SWI/SNF chromatin remodeling subunits to regulate methylation H2AR3 and H4R3 and suppress DNA repair gene expression, subsequently resulting in the sensitization to the DNA damage stress.
3) In cancer section, they could shortly discuss the regulatory interplay between PRMT7 and PRMT5 in regulating cancer metastasis (in the context of breast cancer).
We thank the reviewer for bringing this point to our attention. The following sentence was added:
The allosteric regulation between PRMT7 and PRMT5 should be considered, especially in breast cancer, where the role of PRMT5 in cancer-initiating cells and disease progression has been well established [68, 69].
4) PRMT7 mediates inhibition of cellular senescence through regulation of GLI1 and GLI2 activity. This may be a novel mechanism of PRMT7-mediated tumorigenesis and cancer formation and should be mentioned.
We thank the reviewer for pointing this complex interplay. The following sentence was added to highlight it:
This study is especially important in light of the complex regulation of GLI1 and GLI2 by PRMT1, PRMT5, and PRMT7 that controls cell senescence, self-renewal with potentially far-reaching implications in pluripotency and cancer-initiating cell biology.
5) Another physiological role of PRMT7 is the suppression of adipogenesis through modulation of C/EBP-β activity (DOI: 10.1016/j.bbrc.2019.07.096). The authors could add a small comment on this function.
The following sentence was added, thanks for the suggestion.
Interestingly, PRMT7 also plays a role in adipogenesis by controlling C/EBP-β activity or PPAR-γ2 expression.
Minor points
- Please include affiliation # 2 Corrected
- Line 22: and factors that Corrected
- Line 46: C-terminal domain Corrected
- Line 56: S. cerevisiae Corrected
- Line 97: catalytically active module-N Corrected
- Line 118: PRMT7-directed Corrected
- Line 119: PRMT5-mediated Corrected
- Line 123: may be able Corrected
- Line 148: Selected PRMT7 Corrected
- Line 152: metabolism-associated proteins Corrected
- Line 153: Interestingly, these studies Corrected
- Line 155: proteomic studies Corrected
- Line 368: PRMT7-mediated methylation Corrected
- Line 370: Consequently, the PRMT Corrected

Reviewer 3 Report
The topic is interesting but I think there are some flaws in the review in particular in the goals of the review versus information presented. Maybe there should be a focus on structure and function on PRMT7 in relation to disease models? Introduction is far too short.
The figures are good for the structure section but a wrap-up big picture figure for the function section would be useful. Information given is very bare bones with little author opinion or synthesis of information.
Additionally, a model of PRMT7 cell signaling would be helpful.
- Under table 1 “netherthelness” should be “nevertheless”. There are a few other minor grammatical and spelling errors in the text, such as In line 27, is the intended description for protein arginine methylation “posttranslational modification” rather than “posttranscriptional modification?”. Also, in different parts of the paper, a comma is missing after several transitional words such as interestingly, concurrently, etc.
Author Response
Please use the 2nd response
Round 2
Reviewer 3 Report
The changes are good but they did fullt adderss my suggestions:
1) Their information is disjointed and the introduction could be beefed up a bit. Right now, it is too short.
2) I suggest they need a wrap-up figure of PRMT7 in disease.
3) Some research paper “PRMT7, a New Protein Arginine Methyltransferase That Synthesizes Symmetric Dimethylarginine” by Lee. et al showed PRMT7 catalyzes the formation of SDMA, consequently classifying it as a type II enzyme. This is controversial with the argument the authors proposed. Therefore, it is important to discuss this controversy.
Author Response
Dear Reviewer,
We would like to thank you for your time and helpful comments as well as suggestions. We considered them very carefully, added the requested information, and made changes to the manuscript that will facilitate scientific communication, interpretation and will provide a broad and inspiring overview of PRMT7. Below, as outlined in the following pages, is the detailed list addressing the suggestions and concerns. The changes in the manuscript text are indicated. We hope that the revisions will meet your expectations and look forward to your response.
Yours sincerely,
Dalia
1) Their information is disjointed and the introduction could be beefed up a bit. Right now, it is too short.
We thank the reviewer for this suggestion. We have added the following to provide more information on the PRMT family
PRMTs regulate normal physiological processes such as myogenesis, embryonic development, and immune system function and play roles in pathologies such as cancer, neurodegeneration, and inflammation. Recent knowledge on shared and unique arginine methylated substrates of PRMTs has shed light on the individual members of the PRMT family.
2) I suggest they need a wrap-up figure of PRMT7 in disease.
We thank the reviewer for this suggestion and have carefully considered adding a figure requested. At this point in time, we felt that such a figure would provide only limited benefit as so far PRMT7 has only been extensively investigated in cancer. Most PRMT7 functions have been associated with normal physiological processes, and their relevance to disease would be undue speculation. More research is needed towards that. However, we have added the information on cancer links to Table 1.
3) Some research paper “PRMT7, a New Protein Arginine Methyltransferase That Synthesizes Symmetric Dimethylarginine” by Lee. et al showed PRMT7 catalyzes the formation of SDMA, consequently classifying it as a type II enzyme. This is controversial with the argument the authors proposed. Therefore, it is important to discuss this controversy.
We thank the reviewer for reminding us about this study. The above-mentioned paper was published in 2005 with no follow on studies to support the claims since then. We have added the sentence:
Early studies have reported PRMT7-mediated dimethylation of arginines; however, subsequent evidence on enzymatic activity, structure, and mutagenesis has unequivocally shown monomethylation activity PRMT7.